# RH-BrainFS: Regional Heterogeneous Multimodal Brain Networks Fusion Strategy

**Hongting Ye** [1], **Yalu Zheng** [1], **Yueying Li** [1], **Ke Zhang** [1], **Youyong Kong** [1,2]*, **Yonggui Yuan** [3]
[1]Jiangsu Provincial Joint International Research Laboratory of Medical Information Processing
School of Computer Science and Engineering, Southeast University
[2]Key Laboratory of New Generation Artificial Intelligence Technology and Its
Interdisciplinary Applications (Southeast University), Ministry of Education, China
[3]Department of Psychosomatics and Psychiatry, Zhongda Hospital
School of Medicine, Southeast University
{yehongting, 220212084, 230228504, kylenz, kongyouyong}@seu.edu.cn
yygylh2000@sina.com

## Abstract

Multimodal fusion has become an important research technique in neuroscience that completes downstream tasks by extracting complementary information from multiple modalities. Existing multimodal research on brain networks mainly focuses on two modalities, structural connectivity (SC) and functional connectivity (FC). Recently, extensive literature has shown that the relationship between SC and FC is complex and not a simple one-to-one mapping. The coupling of structure and function at the regional level is heterogeneous. However, all previous studies have neglected the modal regional heterogeneity between SC and FC and fused their representations via "simple patterns", which are inefficient ways of multimodal fusion and affect the overall performance of the model. In this paper, to alleviate the issue of regional heterogeneity of multimodal brain networks, we propose a novel **R**egional **H**eterogeneous multimodal **Brain** networks **F**usion **S**trategy (RH-BrainFS).[2] Briefly, we introduce a brain subgraph networks module to extract regional characteristics of brain networks, and further use a new transformer-based fusion bottleneck module to alleviate the issue of regional heterogeneity between SC and FC. To the best of our knowledge, this is the first paper to explicitly state the issue of structural-functional modal regional heterogeneity and to propose a solution. Extensive experiments demonstrate that the proposed method outperforms several state-of-the-art methods in a variety of neuroscience tasks.

## 1 Introduction

Currently, a large number of neuroscience studies are based on unimodal imaging [2, 30, 47, 56, 61]. However, different brain imaging techniques, such as functional magnetic resonance imaging (fMRI) [54] and diffusion magnetic resonance imaging (dMRI) [44], reflect different aspects of the brain's internal characteristics. Therefore, it is often insufficient to use a single modality of data for neuroscience research and it is necessary to integrate multiple modalities of imaging data to achieve good performance in neuroscience tasks such as depression classification [3, 33] and gender classification [15, 53].

In multimodal brain networks fusion, existing researchs are mainly focused on fusing structural and functional modalities (structural modality is constructed from dMRI and functional modality is

---

*Corresponding author
[2]The codes are available at https://github.com/Yedaxia1/RH-BrainFS.

37th Conference on Neural Information Processing Systems (NeurIPS 2023).

constructed from fMRI) [6, 29, 32, 57, 58]. Most methods directly fuse the two modal representations via "simple patterns" (we define this as **direct interaction**, where two modal features/embeddings are directly combined to perform some computation, e.g. concatenation [32], weighted summation [57], or self-attention [58] techniques.) without considering the issue of regional heterogeneity [37] between this two modalities. However, extensive literature [22, 37] has shown that the relationship between structural connectivity (SC) and functional connectivity (FC) is complex and not a simple one-to-one mapping. Specifically, the coupling of structure and function at the regional level is heterogeneous and follows the molecular, cellular and functional hierarchical structure. In other words, structure may be more tightly coupled to function in some regions than in others. This shows that regional heterogeneity is a key factor in linking different modal brain networks.

Based on the above research gap, in this paper we propose a novel regional heterogeneous multimodal brain networks fusion strategy that aims to fully account for the regional heterogeneity among brain networks and achieve better multimodal brain networks fusion performance.

First, the brain network itself has strong regional characteristics [36, 41], and the combination of neighbouring brain regions can serve as an important criterion for neuroscience tasks [16, 24]. Specifically, regional characteristics behave as subnetwork (also called subgraph) features in the whole brain network [11, 28], yet other multimodal studies do not take this into account. In this paper, we focus on the characteristics of brain regions from the subgraph pattern. The subgraph pattern have become a relatively hot area of graph representation learning research in recent years [25, 39, 55]. Subgraph convolutional networks can obtain potential representations of each subgraph in the graph, that reflect the regional characteristics of the graph [7, 10]. Therefore, to effectively extract the regional characteristics of each brain region in the brain network, we introduce a Brain Subgraph Networks (BrainSubGNN) module in Sec. 3.2. Our BrainSubGNN is divided into two steps, subgraph sampling and subgraph embedding. The former is used to obtain the subgraph partition of the brain network, the latter to aggregate the subgraph characteristics of the brain network.

Next, since several previous studies have shown that the characteristics of different individual regions in the brain network have widely different influences on neuroscience tasks [1, 31, 40, 42], we need to pay attention to the influence of these brain region characteristics on neuroscience tasks and quantify them into accurate values. In this paper, we focus on the Transformer [45] to learn the accurate influence values of these brain region characteristics. Although originally proposed for NLP tasks, there has been recent interest in Transformers [45] as universal perceptual models [14]. Through the attention mechanism, Transformer can learn the accurate influence values of different tokens well for the classification result, which happens to meet our needs.

However, as mentioned above, there is the issue of modal regional heterogeneity between SC and FC, which is a key factor in linking these two modal brain networks [37]. Many previous multimodal fusion methods, via "direct interaction", have not considered this issue between SC and FC, which are inefficient ways of modality fusion and affect the overall performance of the model. Based on that, avoiding direct interaction between two modalities within the Transformer is the aim of our study. Inspired by MBT [26], we present a transformer-based fusion bottleneck (Trans-Bottleneck) module for fusing regional heterogeneous brain networks in Sec. 3.3. Specifically, the Trans-Bottleneck module contains two standard transformers [45] and a certain number of fusion bottlenecks. The standard transformers are used to learn the accurate influence values of each brain region on neuroscience tasks through the attention mechanism. The fusion bottlenecks, as intermediate media for modality fusion, establish connections between regional heterogeneous modalities and learn the key information of each modality in the latent space (we define this as **indirect interaction**, where two modal features/embeddings are not directly combined for the computation.).

The main contributions of this paper are summarized as follows:

- To alleviate the issue of regional heterogeneity of multimodal brain networks, we propose a novel **R**egional **H**eterogeneous multimodal **Brain** networks **F**usion **S**trategy (RH-BrainFS), using BrainSubGNN module and Trans-Bottleneck module to fuse regional heterogeneous multimodal brain networks for neuroscience tasks.
- To the best of our knowledge, this is the first paper to explicitly state the issue of structural-functional modal regional heterogeneity and to propose a solution.
- Extensive experiments demonstrate the effectiveness of RH-BrainFS in multimodal brain networks fusion tasks on depression classification and gender classification datasets.

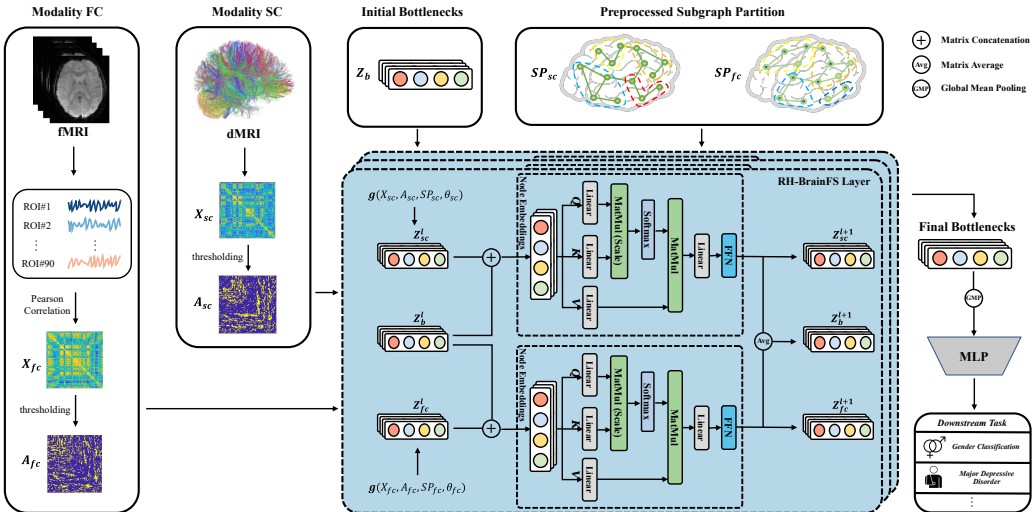

Figure 1: The overall framework of our proposed RH-BrainFS Model. The structural connectivity (SC) and functional connectivity (FC) are constructed from fMRI and dMRI respectively, the initial bottlenecks $Z_b$ are a set of learnable tokens randomly sampled from standard normal distribution, and the subgraph partition ($SP_i$) are obtained in the preprocessing stage. Function $g(\cdot)$ denotes the BrainSubGNN module.

## 2   Related Work

**Brain Subgraph Networks:**  In recent years, subgraph techniques have gained popularity in brain network analysis due to their ability to accurately model certain aspects of brain organization with high consistency to established brain functional systems [27]. This has led to a focus on identifying informative and signaling subgraphs from the entire brain connectome that may be relevant to brain diseases [46].  Various subgraph-based methods have been proposed for brain network research, such as the adaptive dense subgraph discovery (ADSD) model [51] which uses a likelihood-based approach to extract disease-associated subgraphs from group-level whole-brain connectome data. Other methods, such as an earlier approach from [4], balance topological information of local and global graphs using subgraphs, while approach [5] utilizes subgraphs to represent local features in large-scale brain network. Recent research [19] has also classified brain network by extracting contrastive subgraphs, and the SBLR model [49] suggests that subgraphs have attractive neurological interpretations and may correspond to outcome-related anatomical circuits. In this paper, our model builds upon these approaches and employs a BrainSubGNN module to efficiently aggregate regional characteristics of brain network.

**MultiModal Brain Networks Fusion:**  Over the past few years, several methods have been developed for multimodal fusion in neuroscience research [12, 8, 38, 23, 60]. One traditional approach, SNF [48], creates an initial similarity network for each feature and iteratively combines them with a non-linear graph fusion formula to generate a final fusion network. However, recent advancements in deep learning have led to the development of more sophisticated multimodal fusion techniques. For instance, the GBDM [57] model employs both structural and functional information from diffusion and functional magnetic resonance imaging (MRI), respectively, to effectively differentiate individuals with mild cognitive impairment (MCI) from age-matched controls. The MGCN [29] model uses manifold-based regularization terms to account for inter-modality and intra-modality relationships. One approach [13] involves deep collaborative learning to capture cross-modal associations and trait-related features.  Another approach [32] combines representations to fuse multimodal data. Moreover, a method [62] applies a multimodal non-Euclidean brain network analysis technique based on community detection and convolutional autoencoder for epilepsy classification. Additionally, a new adversarial learning-based node-edge graph attention network (AL-NEGAT [6]) has been proposed for autism spectrum disorder (ASD) recognition based on multi-modal MRI data. However, none of these methods have addressed the issue of modal regional heterogeneity [37] in multimodal

brain networks. To fill this gap, we propose a new multimodal brain networks fusion strategy aimed at alleviating the modal regional heterogeneity issue and achieving better fusion performance.

## 3    Method

In this section, we present our proposed regional heterogeneous multimodal brain networks fusion strategy RH-BrainFS (as shown in Fig. 1). We begin by discussing the definition of the multimodal brain networks fusion task in Sec. 3.1. We then explain how the BrainSubGNN module captures the regional characteristics of brain networks in Sec. 3.2. Finally, we describe how the Trans-Bottleneck module used in the RH-BrainFS model alleviate the issue of regional heterogeneity in multimodal brain networks in Sec. 3.3.

### 3.1    Perliminaries

**Multimodal Brain Networks Fusion Task:** Given a multimodal brain networks dataset $D = \{Sample_0, Sample_1, ..., Sample_{L-1}\}$, where each sample represents a person, and $L$ is the size of this dataset. Each $Sample_i = (\mathcal{G}_{sc}, \mathcal{G}_{fc}, y)$ contains a structural connectivity graph (each brain region is viewed as a node in the graph), a functional connectivity graph (ditto) and a class label $y \in \{0, 1\}$ (0, 1 represent different meanings on different tasks). Each kind of input graph $\mathcal{G} = (A, X, V)$ is composed of a node set $V$, an adjacency matrix $A \in \mathbb{R}^{N \times N}$ and node features $X \in \mathbb{R}^{N \times d}$, where $N = |V|$ denotes the number of nodes and $d$ denotes the input dimensional of node features. In the brain network graph, node set $V$ denotes the collection of brain regions, adjacency matrix $A$ denotes the connectivity between various brain regions and node features $X$ denotes the original features of each brain region. In the multimodal brain networks classification task, the purpose is to find a mapping function $g : (\mathcal{G}_{sc}, \mathcal{G}_{fc}) \rightarrow y$.

### 3.2    Brain Subgraph Networks

We now explain how BrainSubGNN module captures the regional characteristics of brain networks. As shown in Fig. 2, the BrainSubGNN contains subgraph sampling step and subgraph embedding step.

#### 3.2.1    Subgraph sampling

The purpose of subgraph sampling is to construct a receptive field for each brain region (also named node in graph), it represent the regional characteristics of this brain region. In our RH-BrainFS, rooted subgraph [59] is utilized to construct receptive field, as the rooted subgraph can exhibit even greater discriminative power than the first-order Weisfeiler-Leman (1-WL) test due to interconnectivity among neighboring nodes [50, 59].

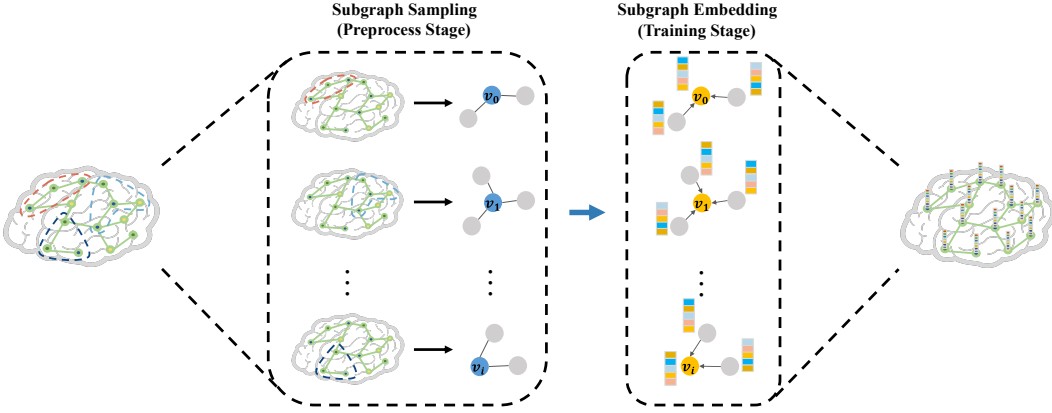

Figure 2: Brain Subgraph Networks (BrainSubGNN). Including subgrah sampling process (1-hop) and subgraph embedding process.

Specifically, we define central subgraph for each node in the whole brain network. The central subgraph describes the surrounding region of the central node. For a input graph $\mathcal{G}$ and the central node $v_i$, the central subgraph is denoted as $\tilde{\mathcal{G}}_i^k = (\tilde{A}_i, \tilde{X}_i, \tilde{V}_i) \subset \mathcal{G}$ where $\tilde{V}_i = \mathcal{N}^k(i)$ contains k-hop neighbor nodes of $v_i$ and $\tilde{A}_i$, $\tilde{X}_i$ equal to the corresponding part in the original $A$ and $X$. The central subgraph $\tilde{\mathcal{G}}_i^k$ constructed above includes brain region $i$ and its k-hops adjacent brain regions. Such a subgraph contains the characteristics of this local brain network, which is of great significance to the brain network. In this work, the subgraph sampling is a preprocessing process before network training, node and edge indice are saved as a binary file. In the training stage, the indice are loaded directly without additional cost on training.

### 3.2.2 Subgraph embedding

The subgraph partitions of the brain network are obtained in the previous step, each subgraph contains a central brain region and several adjacent brain regions, so we expect to be able to use a method that integrates the characteristics of all the brain regions in the subgraph and extract an embedding to represent the entire subgraph. Inspired by the graph isomorphism network [52], we consider the subgraph representation task as a multi-set problem and use multi-layer perceptrons to learn an injective function for aggregating regional characteristics the brain subgraph network effectively and learn subgraph representation as:

$$z_i^{(l)} = \text{MLP}^{(l)} \left( W_1^{(l)} \tilde{h}_i^{(l)} + \sum_{j \in \tilde{V}_i \setminus i} W_2^{(l)} \tilde{h}_j^{(l)} \right) \tag{1}$$

where $\tilde{h}_i$ denote the hidden representation of node $i$, and $W_1^{(l)}$, $W_2^{(l)} \in \mathbb{R}^{d^{(l)} \times d^{(l+1)}}$ are learnable weight matrices for the central node and other nodes in subgraph, respectively. By default, we set the MLP as a two-layer fully connected module. The obtained $z_i$ represents the local characteristics of brain region $i$.

## 3.3 Transformer-Based Fusion Bottleneck

Through the BrainSubGNN described earlier, we obtain the regional representations of each modality of the brain network $Z_i \in \mathbb{R}^{N \times d_{hid}}$. Next, our goal is to apply an efficient fusion strategy to fuse critical and complementary information from $Z_{sc}$ and $Z_{fc}$, extract a distinguishing embedding to represent the features of the whole brain, and serve as a criterion for downstream tasks. We now describe how Trans-Bottleneck module alleviate the issue of regional heterogeneity in multimodal brain networks.

### 3.3.1 Fusion Bottlenecks

Due to the issue of regional heterogeneity between SC and FC, we are committed to avoiding the direct interaction of two modalities, and we prefer to find an intermediate element as a bridge for the interaction between two modalities (means indirect interaction). Inspired by MBT [26], we introduce the fusion bottlenecks into neuroscience research.

Specifically, fusion bottlenecks are simply a set of learnable tokens $Z_b \in \mathbb{R}^{N_b \times d_{hid}}$, where $N_b$ is a hyperparameter denoting the number of fusion bottlenecks. In this work, we utilize the fusion bottlenecks as intermediate medium to bridge contact of this two modalities (SC and FC). As shown in Fig. 3, the fusion bottlenecks allow information to flow between modalities and fusion bottlenecks (indirect interaction), but limit the flow of information between modalities (direct interaction). This procedure avoid direct interaction between regional heterogeneous modality data, effectively improving the model's performance. The initial bottlenecks are a set of learn-

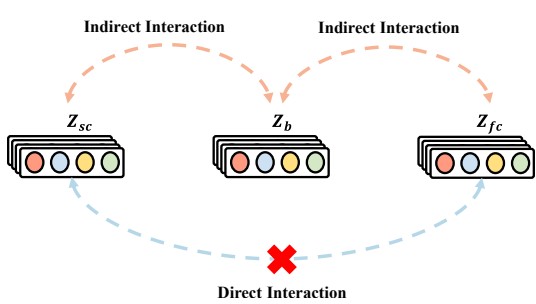

Figure 3: Interaction between three main tokens.

able tokens that are randomly sampled from standard normal distribution $\mathcal{N}(0, 1)$ and then passed to the first RH-BrainFS layer. Finally, the fusion bottlenecks output by the last RH-BrainFS layer are used as the classification basis for downstream tasks (gender classification, major depressive disorder diagnosis, etc.).

### 3.3.2 Transformer-Based Fusion

Although originally proposed for NLP tasks, there has been recent interest in Transformers as universal perceptual models due to their ability to model dense correlations between tokens. Meanwhile, it turns out that individual regional characteristics of brain networks have different influence values for neuroscience tasks [1, 31, 40, 42]. Based on these, our RH-BrainFS method utilizes the Transformer [45] as a baseline for the fusion strategy to capture key subgraph characteristics in brain networks.

Specifically, as shown in Fig. 1, we first concatenate the fusion bottlenecks $Z_b$ and each modality representation $Z_i$, and then feed the concatenated tokens to the standard Transformer [45] model of the corresponding modality. In the Transformer model, the fusion bottlenecks learn important regional characteristics of the brain network for each modality through attention mechanism. The tokens output by the Transformer are re-split according to the previous splicing scheme to obtain new potential features $Z_i^{l+1}$ of the brain network and temporary fusion bottlenecks $\hat{Z}_{b_i}^{l+1}$ corresponding to each modality. After that, the temporary fusion bottlenecks $\hat{Z}_{b_i}^{l+1}$ of all modalities are matrix-averaged (also named average pooling) to obtain the fusion bottleneck latent representation $Z_b^{l+1}$.

This procedure can be formulated as:

$$\left[ Z_i^{l+1} \,||\, \hat{Z}_{b_i}^{l+1} \right] = \text{Transformer}(\left[ Z_i^l \,||\, Z_b^l \right]; \theta_i) \tag{2}$$

$$Z_b^{l+1} = \text{AVERAGE}(\hat{Z}_{b_i}^{l+1}) \tag{3}$$

where $i \in \{SC, FC\}$, $\theta_i$ denotes the modality specific Transformer, $[\cdot \,||\, \cdot]$ denotes the concatenate operation, $Z_b^l$ denotes the fusion bottlenecks in $l^{th}$ layer, and $\hat{Z}_{b_i}^{l+1}$ denotes the temporary fusion bottlenecks in $i^{th}$ modality specific Transformer.

In this procedure, each modality transfers importance regional brain network characteristics within its modality to the fusion bottlenecks, and the fusion bottlenecks utilize shared characteristics between modalities to guide the learning of each modality's brain network in the next layer.

### 3.3.3 Downstream Tasks

In the final stage, we complete classification basis for downstream tasks through the output $Z_b^l$ in the last layer. Specifically, we first use global mean pooling as the readout function, and then input the result into a MLP to complete classification basis.

$$Logits = \text{Softmax}(\text{MLP}(\frac{1}{N_b} \sum_{N_b}^{i} Z_{i,b})) \tag{4}$$

where $Z_{i,b}$ denotes the i-th row of $Z_b$, and $Logits \in \{0, 1\}$. In this paper, on the depression classification task, 0 represents major depressive disorder, 1 represents normal control. And on the gender classification task, 0 represents male, 1 represents female.

## 4 Experiments

In this section, we perform a series of experiments to evaluate the effectiveness of the proposed RH-BrainFS method. First, we provide the detailed experimental settings in Sec. 4.1. Then, we perform comparison experiments on all datasets to compare the performance of different methods in Sec. 4.2. Finally, we perform some ablation studies of the main modules and hyperparameters in the proposed RH-BrainFS method in Sec. 4.3.

## 4.1 Experimental Settings

**Datasets.** We evaluate our RH-BrainFS method on two different classification tasks investigating structure-function fusion. 1) The gender classification task on Human Connectome Project (HCP) dataset [43], which contains 560 female samples and 479 male samples. 2) The Major Depressive Disorder (MDD) diagnosis task on the hospital datasets [17, 18], including the Affiliated Zhongda Hospital of Southeast University (Zhongda hospital) and the Second Affiliated Hospital of Xinxiang Medical University (Xinxiang hospital). This study included 48 controls and 62 MDD patients from the Zhongda hospital and 46 controls and 31 MDD patients from the Xinxiang hospital. We also combine Zhongda and Xinxiang as Two-site dataset to construct more difficult and more realistic tasks.

**Preprocessing.** Here, we would briefly introduce how to construct the brain network of SC from dMRI and FC from fMRI.

- SC. The dMRI data is preprocessed using the brain's diffusion toolbox of FMRIB Software Library [34]. Next we construct $\mathcal{G}_{sc} = (A_{sc}, X_{sc}, V_{sc})$ from preprocessed dMRI data. First, we obtain the brain regions ($V_{sc}$) of the individual space by mapping the anatomical automatic labeling (AAL) template in the standard space to the individual space. Then, we using DSI Studio software [9] to implement Fiber tracking. Finally, we obtain the feature $X_{sc} \in \mathbb{R}^{|V_{sc}| \times |V_{sc}|}$ by counting the number of structural connective fibres between the different regions of the AAL, and the adjacency matrix $A_{sc}$ of the structural graph is obtained by thresholding $X_{sc}$ with a threshold.

- FC. The fMRI data is preprocessed using the Data Processing Assistant for Resting-State Function (DPARSF) [35] MRI toolkit. Next we construct $\mathcal{G}_{fc} = (A_{fc}, X_{fc}, V_{fc})$ from preprocessed fMRI data. First, averaged time series are first computed for each brain region with a predefined atlas. Then, the Pearson correlation is utilized to calculate the functional matrix. Finally, the functional matrix is thresholded by proportional quantization to obtain the adjacency matrix $A_{fc}$. The features $X_{fc}$ are functional connectivity matrix obtained earlier.

**Metrics.** In this study, we evaluate all the methods using 10-fold cross-validation with the same partition of training and testing splits. Our evaluation metrics include classification accuracy (ACC), sensitivity (SEN), specificity (SPE), f1 score (F1) and ROC-AUC (AUC). Higher values for all metrics indicate better performance. We record the mean and standard deviation on 10 random runs with 10-fold cross-validation.

**Implementation Details.** For all experiments, we adopt Adam as the optimizer and StepLR (step_size=50, gamma=0.8) as the scheduler. The initial learning rate is set to 5e-4 and the dropout rate is set to 0.3. Also we utilize a early stop mechanism that 300 epochs patience in total 500 epochs. In the RH-BrainFS model, we set the k-hop in the subgraph sampling to 1, the number of bottlenecks $N_b$ to 4, the number of attention heads in the Transformer to 4, and the total number of network layers to 2. All our experiments are implemented in PyTorch and trained on one NVIDIA 3090.

## 4.2 Comparison Experiments

In this section, we verify the performance of our RH-BrainFS against existing baselines on several datasets.

**Baselines.** We choose two categories of methods as comparison methods, both of which are methods for the direct study of SC and FC. The first category is unimodal methods, including FGDN [20] and BrainGNN [21], where FGDN uses a spectral graph convolution method to extract brain networks features, and BrainGNN proposes a ROI-aware graph convolution layer and uses pooling to extract brain networks features. In our experiments, SC and FC are used as inputs to the unimodal method, respectively. The second category is multimodal methods, including SVM, Random Forest, MGCN [29], GBDM [57], MMGNN [32] and AL-NEGAT [6], where SVM and Random Forest concatenate SC and FC as input, MGCN uses manifold-based regularization terms to consider inter-modality and intra-modality relationships, GBDM adopts weighted summation pattern to fuse multimodal brain networks, MMGNN utilises the concatenation method in multimodal tasks and AL-NEGAT combines multimodal information to construct edge feature maps and node feature maps. Code implementations of all baseline methods are taken from their respective original papers.

Table 1: Comparison experiments results (in percentage) on the all chosen datasets (only the accuracy is shown, the full results can be referred to the Appendix A). The best results are marked in bold. The suboptimal results are marked underlined.

| Method | Modality | Datasets | | | |
|---|---|---|---|---|---|
| | | HCP | Zhongda | Xinxiang | Two-site |
| FGDN | FC | 67.56±3.02 | 65.67±3.26 | 67.91±3.27 | 59.34±2.78 |
| FGDN | SC | 63.42±4.79 | 64.02±3.49 | 65.89±5.15 | 68.91±2.53 |
| BrainGNN | FC | 66.41±6.44 | 69.18±3.39 | 73.46±4.33 | 69.55±3.23 |
| BrainGNN | SC | 67.37±5.89 | 70.73±2.07 | 73.66±3.60 | 69.51±2.58 |
| SVM | SC,FC | 74.49±2.97 | 63.21±2.09 | 71.73±1.99 | 66.06±1.56 |
| Random Forest | SC,FC | 68.24±2.94 | 61.45±2.80 | 62.78±1.63 | 62.43±2.19 |
| MGCN | SC,FC | 67.94±5.41 | 75.18±2.34 | 82.24±3.71 | 72.98±2.17 |
| GBDM | SC,FC | 71.02±4.39 | 74.81±2.44 | 80.71±2.83 | 72.48±1.91 |
| MMGNN | SC,FC | 73.33±2.82 | 60.69±3.61 | 68.21±4.44 | 59.72±3.18 |
| AL-NEGAT | SC,FC | 75.12±3.66 | 73.95±3.45 | 75.75±3.81 | 71.86±2.49 |
| RH-BrainFS (ours) | SC,FC | **78.63±4.36** | **80.64±1.58** | **90.27±2.00** | **78.48±1.43** |

**Results.** As shown in Tab. 1, our model significantly outperforms the other comparison methods on the all selected datasets. The results show that multimodal methods generally have better performance than unimodal methods because they capture more complementary information. Among the multimodal methods, our RH-BrainFS method achieves the best performance on the all the selected datasets (improvement of 3.51% on HCP, 5.46% on Zhongda hospital dataset, 8.03% on Xinxiang hospital dataset and 5.50% on two-site dataset). The reason for the performance improvement is that our RH-BrainFS method fully considers the issue of regional heterogeneity between SC and FC and proposes an appropriate solution strategy for this issue.

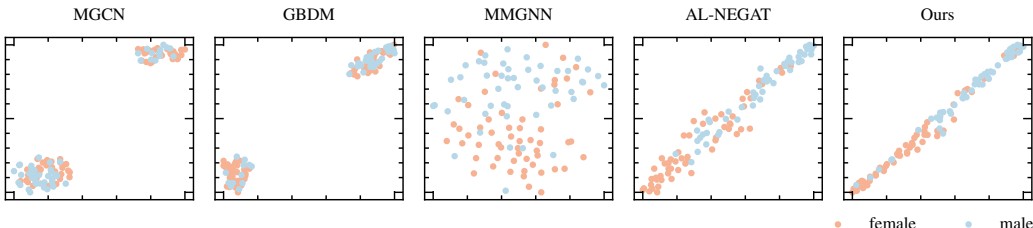

Figure 4: t-SNE visualisation of multimodal method on HCP dataset. Each dot denotes a test sample.

**Visualization.** In order to intuitively display the performance of each multimodal method, we conduct visualization experiments on the HCP dataset. Specifically, we take the graph-level embedding output from the last layer of each multimodal method for t-SNE visualisation. As shown in Fig. 4, although the MGCN and GBDM all form two clusters, these two clusters do not distinguish the two types of samples well and there is still a lot of confusion. The MMGNN is loosely distributed and does not form good class boundary. It can be seen that the Al-NEGAT produces a similar distribution to our method, but there's still a lot of confusion at the class boundary. In contrast, our method eliminates the confusion at the class boundary, achieves a good classification effect, most of the samples can be accurately distinguished and only a small number of samples have errors.

## 4.3 Ablation Study

In this section, we further perform ablation studies on the main modules and hyperparameters in the proposed RH-BrainFS method.[3]

---

[3]Full results of hyperparameter experiments refer to Appendix B.

Table 2: Ablation study of the main modules in the RH-BrainFS method. The bold font indicates that the evaluation metric achieves the best performance in the ablation study.

| Modules | ACC | SEN | SPE | F1 | AUC |
|---|---|---|---|---|---|
| w/o BrainSubGNN Trans-Bottleneck | 76.23±3.78 | 71.40±11.94 | 80.36±7.70 | 73.00±6.27 | 75.88±4.12 |
| w/o BrainSubGNN | 77.22±3.89 | 68.44±12.16 | **82.86±8.61** | 72.16±6.34 | 75.65±4.19 |
| w/o Trans-Bottleneck | 76.80±2.93 | 71.14±10.50 | 81.61±7.10 | 73.50±5.13 | 76.38±3.27 |
| RH-BrainFS (ours) | **78.63±4.36** | **75.59±6.75** | 81.25±6.04 | **76.49±4.91** | **78.42±4.38** |

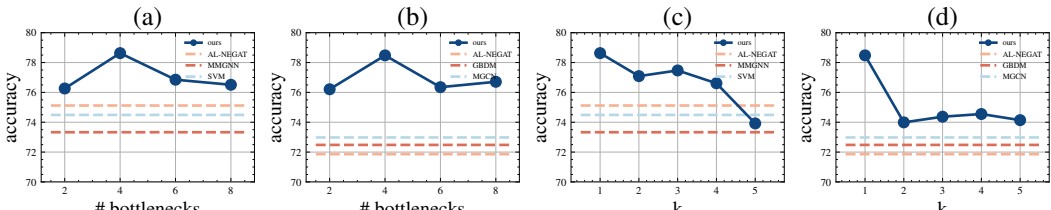

Figure 5: The effect of varying hyperparameters. (a) and (c) are experiments on HCP datasets, (b) and (d) are experiments on Two-site datasets.

**Effectiveness of Main Modules.** First, we perform an ablation study on HCP dataset to validate effectiveness of the main modules, BrainSubGNN and Trans-Bottleneck. Specifically, we replace BrainSubGNN with normal GIN [52] (process on the whole brain network), and replace Trans-Bottleneck with a standard transformer (compute self-attention directly between two modalities, equal to direct interaction), respectively. As shown in Tab. 2, we can find that both BrainSubGNN and Trans-Bottleneck have a certain effect on the performance of the model (compare the first three rows in the table). Furthermore, we find that the two modules are compatible and complementary, as our RH-BrainFS method (combining the two modules) achieves the best performance among the ablation study.

**Impact of Bottlenecks Number.** We then investigate the impact of varying bottlenecks number. Specifically, we run experiments with # bottlenecks=2,4,6,8, respectively, with all other parameters unchanged. In order to ensure the credibility of the experiment, we perform experiments on two datasets (HCP dataset and Two-site dataset). As shown in Fig. 5, on both datasets, the performance of any number of bottlenecks exceeds the baselines. The main reason for this results is that the Trans-Bottleneck module takes into account the regional heterogeneity issue between SC and FC, and avoids the direct interaction of heterogeneous information, thus achieving good fusion performance. Such results demonstrate the importance of the regional heterogeneity issue in the modality fusion process of SC and FC. At the same time, we find that when # bottlenecks=4, the model performance reaches the best, which reflects that only a small number of bottlenecks is needed to achieve a good multi-modal fusion expression capability.

**Impact of Subgraph Sampling Hops.** Next, we investigate the impact of varying sampling hops of subgraph. In this experiment, we set the range of sampling hops of subgraph from 1 to 5. Likewise, we run experiments on HCP dataset and Two-site dataset. As shown in Fig. 5, it can clearly be seen that as the number of sampling hops increases, the performance of the model generally shows a downward trend, and when the sampling hops are too large, the model actually performs worse than the baselines. The reason for this results is that the brain network itself has strong regional characteristics, and too large sampling hops will cause the sampled subgraph to lose the local characteristics of the brain network and tend towards global characteristics.

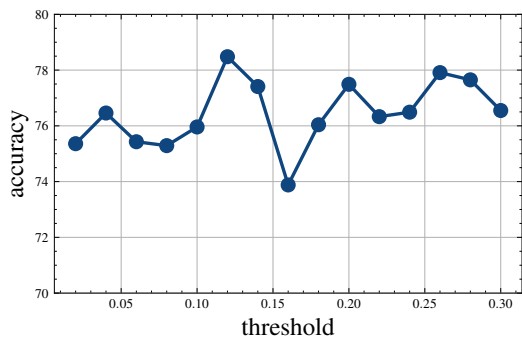

Figure 6: The effect of varying thresholding values.

The results show that only 1 hop of subgraph sampling is needed to express the local characteristics of the brain network well.

**Impact of Thresholding Values.** Finally, in this paper, thresholding is used to obtain the adjacency matrix of the two modalities during data processing (thresholding $X_{sc}$ and $X_{fc}$ to obtain $A_{sc}$ and $A_{fc}$, respectively). Thresholding value has been an important ablation study in neuroscience, and the same experiment was conducted in this paper. As shown in Fig. 6, we set the value range of thresholding to [0.02, 0.30], with 0.02 as a step, for a total of 15 thresholding experiments. From the experimental results, there is no clear trend between threshold values and model performance, but the model performs poorly when the threshold value is too high or too low, so in this paper we chose a threshold value of 0.12 as this is when the model performs best.

## 5   Discussions

**Conclusion.** In this paper, we introduce a brain subgraph networks and a transformer-based fusion bottleneck to alleviate the issue of regional heterogeneity between SC and FC, and propose a novel multimodal brain networks fusion strategy (RH-BrainFS). To the best of our knowledge, this is the first paper to explicitly state the issue of structural-functional modal regional heterogeneity and to propose a solution. We validate our method on a variety of downstream task datasets, achieving state-of-the-art performance.

**Limitations and Future Work.** Due to the scarcity and difficulty of collecting and processing data in neuroscience, the only datasets currently available are very limited, despite the fact that we have spent a great deal of manual effort in this area, and thus the research in this paper suffers from a number of possible data bias issues. In our future work, on the one hand, we will do more work on data to mitigate the problem of data bias. On the other hand, although this paper proposes the use of indirect rather than direct interaction, the two may not be mutually exclusive, and in the future we will investigate how to combine the two to achieve better research results.

**Ethical Issues.** With regard to possible ethical issues in data collection, the Human Connectome Project (HCP) dataset, as a publicly available dataset that has been used in numerous previous studies, is undoubtedly not ethically questionable. It is true that the hospital dataset is held in collaboration with our partner hospitals and is not yet publicly available, but the data is collected with the consent of the subjects who are clearly informed of the purpose of the sample collection, and all identifying information about the sample is hidden in the hospital dataset. Therefore, it does not adversely affect any individual, so there are no ethical or moral issues.

**Possible Negative Social Impacts.** As the research in this paper deals with the diagnosis of depression, it is necessary to elaborate here on the possible negative social impacts of this work, despite the fact that all the current work is at the stage of scientific research and has not been put to practical use. Including but not limited to:

- Incorrect diagnosis. AI methods must have the possibility of error, which cannot be avoided, but an incorrect diagnosis will have a significant impact on individuals and society. Therefore, AI tools can only be used as a diagnostic aid, not as a decision maker, and the final decision should still be made by the doctor.
- Leakage of privacy information. In depression dataset, the identity information of the subjects is highly private, and the leakage of identity information will also have unpredictable and significant impact on individuals and society. Therefore, in this work, we have completely hidden the subjects' identifying information (which is also not visible to the staff in the study group) as a way of preventing the leakage of private information.

## Acknowledgments and Disclosure of Funding

This work is supported by 2022YFE0116700 National Key Research and Development Program of China, supported by 82271570, 31800825 and 31640028 National Natural Science Foundation of China, and also supported by the Big Data Computing Center of Southeast University. This work is partly supported by grant 2242023k30052 Central University Basic Research Fund of China. The authors declare that they have no known competing financial interests or personal relationships that could have appeared to influence the work reported in this paper.

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

# Appendix A: Full Results of Comparison Experiments

Due to the layout of the text and page limitations, only the experimental results for the ACC metric are presented in the main text. However, to ensure a comprehensive presentation of the results, the full results of the comparison experiments involving all four datasets, HCP, Zhongda Hospital, Xinxiang Hospital and Two-site, are presented in Tables 1 to 4. Four tables are presented detailing the mean and standard deviation values for each evaluation metric. The most exceptional results are shown in bold, while results below the optimal threshold are underlined for clarity and emphasis.

**Analysis.** Based on the results from the four tables, our RH-BrainFS method achieved excellent results in all four datasets. Specifically, we achieved the best results on three metrics in the HCP dataset, four metrics in both the Zhongda and Xinxiang Hospital datasets, and an impressive five metrics in the Two-site dataset. This demonstrates the strong performance of RH-BrainFS in effectively completing the fusion classification task for multimodal brain networks. Furthermore, we observe that on the hospital dataset, our RH-BrainFS method significantly outperforms most other methods in terms of bias in each metric, indicating higher stability. This improved stability contributes to the reliability and robustness of RH-BrainFS in fusion classification tasks.

Table 3: Comparison experiments results on the HCP dataset.

| Method | Modality | HCP Dataset | | | | |
|---|---|---|---|---|---|---|
| | | ACC | SEN | SPE | F1 | AUC |
| FGDN | FC | 67.56±3.02 | 52.37±12.55 | 80.54±8.24 | 59.09±6.98 | 66.45±3.49 |
| FGDN | SC | 63.42±4.79 | 53.98±20.69 | 71.43±13.72 | 55.12±14.65 | 62.71±5.57 |
| BrainGNN | FC | 66.41±6.44 | 68.92±6.92 | 63.66±12.30 | 65.41±5.15 | 66.29±6.04 |
| BrainGNN | SC | 67.37±5.89 | 68.39±6.07 | 66.17±8.50 | 65.88±4.45 | 67.28±5.80 |
| SVM | SC,FC | 74.49±2.97 | 71.18±4.95 | 77.32±3.30 | 71.96±3.49 | 74.25±3.05 |
| Random Forest | SC,FC | 68.24±2.94 | 54.88±5.78 | 79.64±4.16 | 61.31±4.44 | 67.26±3.04 |
| MGCN | SC,FC | 67.94±5.41 | 74.53±8.20 | 62.32±7.76 | 68.10±5.59 | 68.42±5.41 |
| GBDM | SC,FC | 71.02±4.39 | 61.95±12.37 | 79.18±8.58 | 65.76±6.35 | 70.56±4.32 |
| MMGNN | SC,FC | 73.33±2.82 | 71.17±4.52 | 75.17±5.67 | 71.10±2.88 | 73.17±2.72 |
| AL-NEGAT | SC,FC | 75.12±3.66 | 72.86±7.74 | **84.46±5.05** | 76.13±4.70 | **78.66±3.81** |
| RH-BrainFS (ours) | SC,FC | **78.63±4.36** | **75.59±6.75** | 81.25±6.04 | **76.49±4.91** | 78.42±4.38 |

Table 4: Comparison experiments results on the Zhongda hospital dataset.

| Method | Modality | Zhongda Dataset | | | | |
|---|---|---|---|---|---|---|
| | | ACC | SEN | SPE | F1 | AUC |
| FGDN | FC | 65.67±3.26 | 78.31±8.26 | 49.25±12.11 | 70.10±4.21 | 63.78±3.86 |
| FGDN | SC | 64.02±3.49 | 65.67±14.37 | 61.50±16.65 | 60.67±8.24 | 63.58±3.56 |
| BrainGNN | FC | 69.18±3.39 | 73.10±5.23 | 64.05±5.00 | 72.05±3.61 | 68.57±3.46 |
| BrainGNN | SC | 70.73±2.07 | 75.93±3.93 | 64.10±3.58 | 73.81±2.33 | 70.01±2.04 |
| SVM | SC,FC | 63.21±2.09 | 74.17±2.06 | 49.00±3.22 | 68.87±2.07 | 61.58±2.20 |
| Random Forest | SC,FC | 61.45±2.80 | 86.69±2.78 | 29.00±4.28 | 71.53±2.17 | 57.85±2.96 |
| MGCN | SC,FC | 75.18±2.34 | 88.52±3.99 | 57.10±8.06 | 80.15±1.91 | 72.81±2.96 |
| GBDM | SC,FC | 74.81±2.44 | 81.77±5.21 | 58.14±8.37 | 74.92±3.20 | 69.96±3.11 |
| MMGNN | SC,FC | 60.69±3.61 | 70.35±9.19 | 47.90±9.32 | 64.45±6.48 | 59.12±3.50 |
| AL-NEGAT | SC,FC | 73.95±3.45 | **90.71±7.24** | 52.05±7.69 | 79.00±4.79 | 71.38±3.54 |
| RH-BrainFS (ours) | SC,FC | **80.64±1.58** | 90.05±5.58 | **68.45±9.32** | **83.96±1.13** | **79.25±2.24** |

Table 5: Comparison experiments results on the Xinxiang hospital dataset.

| Method | Modality | Xinxiang Dataset | | | | |
|---|---|---|---|---|---|---|
| | | ACC | SEN | SPE | F1 | AUC |
| FGDN | FC | 67.91±3.27 | 43.58±8.18 | 84.50±8.30 | 42.03±6.59 | 64.04±2.88 |
| FGDN | SC | 65.89±5.15 | 61.83±14.81 | 68.85±15.06 | 53.03±8.16 | 65.34±3.82 |
| BrainGNN | FC | 73.46±4.33 | 54.92±10.57 | 86.25±6.10 | 55.72±9.30 | 70.58±4.88 |
| BrainGNN | SC | 73.66±3.60 | 56.83±10.76 | 85.35±8.26 | 57.39±8.58 | 71.09±3.76 |
| SVM | SC,FC | 71.73±1.99 | 49.92±4.35 | 86.55±3.42 | 54.52±4.72 | 68.23±2.19 |
| Random Forest | SC,FC | 62.78±1.63 | 12.08±4.81 | **97.20±1.96** | 17.03±6.61 | 54.64±2.11 |
| MGCN | SC,FC | 82.24±3.71 | 74.16±6.62 | 87.45±3.45 | 76.15±5.88 | 80.80±4.15 |
| GBDM | SC,FC | 80.71±2.83 | 60.20±10.03 | 88.43±5.86 | 63.46±9.11 | 73.31±5.83 |
| MMGNN | SC,FC | 68.21±4.44 | 57.33±6.35 | 75.10±8.12 | 55.48±6.03 | 66.21±4.06 |
| AL-NEGAT | SC,FC | 75.75±3.81 | 56.42±14.80 | 88.50±6.35 | 61.19±10.80 | 72.46±5.34 |
| RH-BrainFS (ours) | SC,FC | **90.27±2.00** | **80.75±5.86** | 96.65±2.34 | **85.43±3.90** | **88.70±2.48** |

Table 6: Comparison experiments results on the Two-site dataset.

| Method | Modality | Two-site Dataset | | | | |
|---|---|---|---|---|---|---|
| | | ACC | SEN | SPE | F1 | AUC |
| FGDN | FC | 59.34±2.78 | 54.70±10.82 | 63.79±8.83 | 49.96±8.04 | 59.24±2.85 |
| FGDN | SC | 68.91±2.53 | 70.02±12.42 | 67.83±9.78 | 66.31±7.03 | 68.93±2.52 |
| BrainGNN | FC | 69.55±3.23 | 69.62±5.23 | 68.36±6.67 | 67.54±3.34 | 68.99±3.22 |
| BrainGNN | SC | 69.51±2.58 | 68.12±5.41 | 69.24±6.66 | 66.76±3.42 | 68.68±3.03 |
| SVM | SC,FC | 66.06±1.56 | 58.01±2.49 | 74.04±1.77 | 62.03±2.27 | 66.03±1.56 |
| Random Forest | SC,FC | 62.43±2.19 | 56.17±2.87 | 68.60±3.48 | 59.04±2.73 | 62.38±2.20 |
| MGCN | SC,FC | 72.98±2.17 | 74.02±7.03 | 73.74±4.05 | 72.45±4.48 | 73.88±2.22 |
| GBDM | SC,FC | 72.48±1.91 | 71.53±7.55 | 65.97±5.61 | 68.74±4.92 | 68.75±2.90 |
| MMGNN | SC,FC | 59.72±3.18 | 65.10±4.83 | 54.42±4.42 | 59.96±4.02 | 59.76±3.15 |
| AL-NEGAT | SC,FC | 71.86±2.49 | 75.26±3.62 | 68.12±6.19 | 72.16±2.24 | 71.69±2.56 |
| RH-BrainFS (ours) | SC,FC | **78.48±1.43** | **76.20±4.06** | **80.72±3.60** | **77.35±1.97** | **78.46±1.43** |

# Appendix B: Full Results of Hyperparameter Experiments

In the main text, our hyperparameter experiments are performed on the HCP dataset and the Two-site dataset. We explore two key hyperparameters in RH-BrainFS: the number of bottlenecks and the number of subgraph sampling hops k. However, in the main text we have only presented visualisations of the results of the hyperparameter experiments without giving specific numerical values. Hence, we present here the complete experimental results in Tables 5 to 8.

Table 7: Hyperparameter experiments of varying bottlenecks number on HCP dataset.

| # bottlenecks | HCP Dataset | | | | |
|---|---|---|---|---|---|
| | ACC | SEN | SPE | F1 | AUC |
| 2 | 76.26±3.58 | 69.70±11.51 | 80.00±5.46 | 71.74±6.13 | 74.85±4.06 |
| 4 | **78.63±4.36** | **75.59±6.75** | **81.25±6.04** | **76.49±4.91** | **78.42±4.38** |
| 6 | 76.85±3.41 | 69.72±8.32 | 80.54±7.38 | 72.29±4.26 | 75.13±3.46 |
| 8 | 76.51±3.05 | 71.16±9.82 | 81.25±8.11 | 73.45±4.42 | 76.20±3.18 |

Table 8: Hyperparameter experiments of varying bottlenecks number on Two-site dataset.

| # bottlenecks | Two-site Dataset | | | | |
|---|---|---|---|---|---|
| | ACC | SEN | SPE | F1 | AUC |
| 2 | 76.21±1.17 | 68.88±4.98 | **83.61±4.56** | 73.21±2.33 | 76.24±1.15 |
| 4 | **78.48±1.43** | **76.20±4.06** | 80.72±3.60 | **77.35±1.97** | **78.46±1.43** |
| 6 | 76.35±1.43 | 70.39±3.82 | 82.10±4.18 | 73.80±2.06 | 76.24±1.48 |
| 8 | 76.71±1.32 | 73.68±5.34 | 79.71±4.83 | 74.97±2.28 | 76.69±1.31 |

Table 9: Hyperparameter experiments of varying subgraph sampling hops on HCP dataset.

| k | HCP Dataset | | | | |
|---|---|---|---|---|---|
| | ACC | SEN | SPE | F1 | AUC |
| 1 | **78.63±4.36** | **75.59±6.75** | 81.25±6.04 | **76.49±4.91** | **78.42±4.38** |
| 2 | 77.09±3.53 | 71.35±12.30 | 81.96±9.40 | 73.72±5.73 | 76.66±3.81 |
| 3 | 77.47±3.31 | 74.73±6.83 | 79.82±5.36 | 75.27±4.00 | 77.28±3.39 |
| 4 | 76.61±4.00 | 69.07±11.44 | **83.04±4.01** | 72.63±6.79 | 76.05±4.50 |
| 5 | 73.92±2.94 | 64.50±6.03 | 81.96±6.41 | 69.44±3.56 | 73.23±2.88 |

Table 10: Hyperparameter experiments of varying subgraph sampling hops on Two-site dataset.

| k | Two-site Dataset | | | | |
|---|---|---|---|---|---|
| | ACC | SEN | SPE | F1 | AUC |
| 1 | **78.48±1.43** | **76.20±4.06** | **80.72±3.60** | **77.35±1.97** | **78.46±1.43** |
| 2 | 73.99±1.32 | 69.49±2.88 | 78.42±3.04 | 71.53±2.04 | 73.96±1.32 |
| 3 | 74.37±1.81 | 72.47±5.79 | 76.26±3.42 | 72.66±3.37 | 74.36±1.79 |
| 4 | 74.55±1.64 | 73.13±6.88 | 75.92±6.98 | 72.96±2.68 | 74.53±1.63 |
| 5 | 74.14±1.25 | 70.41±5.73 | 77.79±4.62 | 71.70±3.15 | 74.10±1.28 |

