# OpenReview forum: "RH-BrainFS: Regional Heterogeneous Multimodal Brain Networks Fusion Strategy"
_NeurIPS.cc/2023/Conference — NeurIPS 2023 poster_

### Official Review · Reviewer_kgkZ · 2023-06-16

**Soundness:** 2 fair
**Presentation:** 2 fair
**Contribution:** 2 fair
**Rating:** 5
**Confidence:** 1

**Summary:**

The submission is not in my area, and it's difficult to give reasonable comments about this submission. My research interests focus on medical image reconstruction. please find another appropriate reviewer to review this paper.

**Strengths:**

N/A

**Weaknesses:**

N/A

**Questions:**

N/A

---

### Official Review · Reviewer_UEYi · 2023-07-03

**Soundness:** 3 good
**Presentation:** 3 good
**Contribution:** 2 fair
**Rating:** 3
**Confidence:** 4

**Summary:**

This paper proposes a novel approach called the Regional Heterogeneous Multimodal Brain Networks Fusion Strategy (RH-BrainFS) to address the issue of regional heterogeneity between structural connectivity (SC) and functional connectivity (FC) in brain networks fusion. The proposed approach includes a brain subgraph networks module to extract regional characteristics of brain networks and a transformer-based fusion bottleneck module to alleviate the issue of regional heterogeneity between SC and FC. The authors claim that this is the first work to propose a solution to the issue of structural-functional modal regional heterogeneity. The paper presents extensive experiments that demonstrate that the proposed method outperforms several state-of-the-art methods in a variety of neuroscience tasks.

**Strengths:**

- The paper is well-structured and clearly written. The authors provide a clear introduction to the problem, a thorough review of related work, and a detailed description of their proposed method. They also clearly explain the experimental settings and datasets used in their research. The use of technical terms is appropriate for the intended audience, and the authors provide sufficient context and explanation to make the content understandable.
- The issue of regional heterogeneity in multimodal brain networks is a significant challenge in the field, and the authors' proposed solution has the potential to improve the performance of multimodal fusion models.
- The authors provided detailed preprocessingand implementation parameters.

**Weaknesses:**

- It is important for the authors to compare their proposed approach with existing methods in the field of multimodal brain network modeling. One such method is the joint embedding of structural and functional brain networks with graph neural networks proposed by Zhu et al.. The authors should compare their proposed approach with this method in terms of performance for modality fusion.
- The transformer-based fusion is not original in terms of modality fusion. The authors are suggested to investigate and provide concrete interpretations on the fusion bottlenecks which utilize shared characteristics between modalities.
- Lack of variant on GNN backbone. There are a bunch of GNN architectures for brain network modeling. While the paper acknowledges the use of a BrainSubGNN method, it would be valuable to consider other GNN architectures that have been developed specifically for brain network analysis. By incorporating a broader range of GNN backbones and comparing their performance, the authors can enhance the robustness and generalizability of their proposed approach for multimodal brain network fusion.
- Lack of interpretation analysis. In the context of multimodal brain network modeling, interpretation analysis could involve identifying the brain regions or networks that are most relevant for the classification task, and investigating the functional and structural connectivity patterns that contribute to these regions.

**Questions:**

- How does the proposed approach compare to other state-of-the-art methods in terms of computational efficiency? Is the proposed approach computationally feasible for large-scale datasets?
- The paper mentions that the proposed approach includes a brain subgraph networks module to extract regional characteristics of brain networks. Can the authors provide more details on the design motivation this module, as well as its neurological justification?
- How would the subgraph sampling affect the effectiveness of the overall framework?
- Would there be any potential modality bias in the experimented data?
- Could you elaborate on how woyld the different brain imaging techniques mentioned in the text, such as functional magnetic resonance imaging (fMRI) and diffusion magnetic resonance imaging (dMRI)? reflect different aspects of the brain's internal characteristics?

**Limitations:**

The authors have acknowledged some of the limitations of their work, such as the limited sample size and the difficulty of data collection in neuroscience. However, the paper does not provide a detailed discussion on the technical bias, e.g., modality bias during fusion of the proposed approach, nor does it address potential negative societal impact of the research. Adding those discussions would enhance the paper's credibility and relevance, and help readers better understand the potential implications of the proposed approach.

---

> ### Author Rebuttal · Authors · 2023-08-05
>
> Answers to weaknesses：
> 1. For the first weakness, in fact, we have followed this work (Zhu et al.) , but this work does not disclose their code, and we cannot compare them.
> 2. For the second weakness, the initial motivation for fusion bottleneck was that we believed that there was regional heterogeneity between SC and FC, and thus we believed that their features or potential embeddings would be misaligned, and thus direct interaction would lead to poorer results. Therefore, we proposed indirect interactions for fusion bottlenecks, and later also experimentally verified the effectiveness of indirect interactions (compare the ablation experiments in the third and fourth rows of Table 2). Therefore, we removed the direct interaction between SC and FC (i.e. direct interaction is forbidden) and adopted the indirect interaction of the fusion bottleneck.
> 3. For the third weakness, thanks to this suggestion, we have added this ablation experiment by replacing the BrainSubGNN module with four common GNN frameworks, and the results are shown in the table. (Note: Because the computational efficiency of our model is not as good as other methods, we only chose a simple GNN as the backbone, not a complex GNN).
>
> | Backbone             | Two\-site Dataset |              |              |              |              |
> |:--------------------:|:-----------------:|:------------:|:------------:|:------------:|:------------:|
> |                      | Acc               | Sen          | Spe          | F1           | Auc          |
> | GCN                  | 70\.04±1\.16      | 65\.40±5\.26 | 74\.34±4\.39 | 66\.25±3\.23 | 69\.87±1\.13 |
> | GAT                  | 71\.08±1\.26      | 71\.83±7\.87 | 70\.02±8\.80 | 69\.23±2\.70 | 70\.93±1\.30 |
> | GIN                  | 74\.65±0\.99      | 70\.61±7\.49 | 78\.50±5\.99 | 72\.07±3\.19 | 74\.56±1\.18 |
> | GraphSAGE            | 70\.03±1\.63      | 66\.40±5\.94 | 73\.16±6\.34 | 67\.12±2\.51 | 69\.78±1\.63 |
> | BrainSubGNN \(ours\) | **78\.48±1\.43**      | **76\.20±4\.06** | **80\.72±3\.60** | **77\.35±1\.97** | **78\.46±1\.43** |
>
> 4. For the fourth weakness, thanks for the good advice, we had thought about doing brain region importance analyses, but SC and FC are done separately, meaning that we can only know separately which brain regions are important for SC and which brain regions are important for FC, without an overall result. Secondly, we thought brain region importance analyses were not very relevant to our research motivation, and we were more focussed on solving the problem of regional heterogeneity than on studying brain region importance.
>
> Answers to questions:
>
> 1. For the first question, we supplemented the time-consumption experiment (record the average training time for 100 epoches and the average time for 1 inference on the Two-site dataset), and as we would expect, our method incurred a greater time overhead.
>
>    To explain such results, we believe that firstly MGCN, GBDM and MMGNN are more focussed on researching on processing of data features, so common GNN frameworks (which are simpler compared to Transformer or Attention) are used in the network structure, so they have better good computational efficiency.
>
>    Secondly, we consider AL-NEGAT to be an attentional-based approach that has similarities with our Transformer-based, so AL-NEGAT generates a time overhead that belongs to the same order of magnitude as Ours.
>
>    In response to such results, we admit that our method does underperform in terms of computational efficiency, but falls within an acceptable range, after all, our model performance is better improved.
>
>    Finally, our work can be adapted to large-scale data because each brain network graph is not large (only 90 nodes), so there is no problem of large-scale incomputability.
>
> | **Method**    | **Train Tine \(s / 100 epoches\)** | **Inference Time Cost \(s / 1 inference\)** |
> |:-------------:|:-----------------------------------:|:--------------------------------------------:|
> | **MGCN**      | 23\.875                           | 0\.017  |
> | **GBDM**      | 20\.388                          | 0\.013 |
> | **MMGNN**     | 23\.905                          | 0\.016  |
> | **AL\-NEGAT** | 31\.573                         | 0\.023 |
> | **Ours**      | 33\.875                          | 0\.025  |
>
> 2. For the second question, ss we mentioned inside introduction section (in line 42-43), the brain network itself has strong llocal characteristics, so we used a brain subgraph network to extract local characteristics efficiently.
>
> 3. For the third question, different subgraph sampling strategies extract different brain local characteristics, and different hop counts reflect different brain local characteristics (refer to Fig. 5 of the ablation experiment), as stated in lines 329 to 331 of the paper, too large subgraph sampling hop counts will make the sampled subgraphs not have local characteristics
>
> 4. For the fourth and fifth questions, structural neuroimaging datadiffusion magnetic resonance imaging (dMRI), reflecting voxel tissue density/volume or structural connectivity. The main purpose of structural studies is to reveal anatomical relationships in the brain, which can then be used for prediction.
>
>    Functional neuroimaging datafunctional magnetic resonance imaging (fMRI), reflecting changes in deoxyhemoglobin concentration caused by task-induced or spontaneously regulated neurometabolic. The main purpose of functional studies focuses on dynamic changes in brain activity or connectivity.
>
>    Because of the essential differences, there is necessarily a modal difference between structure and function.
>
> Answers to limitations:
>    1. Thank you for your suggestion, we really should discuss our technical bias in terms of limitations. Regarding the mentioned negative social impacts, we have explained them in detail in the global rebuttal, and in short we can assure you that this work will not have any negative social impacts.

---

> ### Comment · Reviewer_UEYi · 2023-08-21
> **Response to Rebuttal**
>
> Thanks the authors for the response and the added experiments. After reading the rebuttal, I still think compare with existing multimodal methods for brain network modeling is needed. I would like to remain my rating.

---

> > ### Author Response · Authors · 2023-08-21
> >
> > Thank you for all your previous comments.
> >
> > Regarding the issue of the comparison method, I have emailed the authors (Zhu et al.) for the code a long time ago, but did not receive a reply. In the section 4.2, we have compared many existing multimodal methods for brain network modeling (e.g. MGCN, GBDM, MMGNN and AL-NEGAT), and we will describe these more detailed later in the revised version.

---

### Official Review · Reviewer_7Tgh · 2023-07-04

**Soundness:** 3 good
**Presentation:** 4 excellent
**Contribution:** 3 good
**Rating:** 6
**Confidence:** 4

**Summary:**

The paper identifies a gap in the literature of multimodal brain networks fusion in which current methods are said to only use "simple patterns" to fuse modalities, ie, concatenation, weighted summation, and self-attention. To tackle this issue, the paper proposes RH-BrainFS, a new model fusing structural connectivity and functional connectivity data constructed from fMRI and dMRI, respectively. This model includes a BrainSubGNN model for subgraph sampling and subgraph embedding processes, as well as a fusion bottleneck based on transformers.

**Strengths:**

The results of the experiments (as seen in table 1) seem to be particularly strong, and this work is very significant given the lack of varied work in the field of multimodal brain imaging fusion. It is very good that the authors use more traditional ML models beyond deep learning (ie, SVM and random forests) to evaluate the relevance of this work, and that several metrics are considered beyond the simple accuracy metric. Finally, it usually takes a lot of time and manual work to preprocess dMRI data to be in the same parcellation as the fMRI data, thus the fact that this data is not available out-of-the-box clearly adds to the significance of this applied work - I hope the authors will make this preprocessed dataset available at a certain point.


---------------------------------
EDIT AFTER REBUTTAL PERIOD:

From the rebuttal period, the authors did a very good job in addressing most of my concerns, and therefore I'm increasing my decision from borderline accept to weak accept, the soundness score from 2 to 3, and the presentation score from 3 to 4.

**Weaknesses:**

I identify three main weaknesses that require clarification during the rebuttal process and are the reason for me to give a borderline accept despite the strengths. I will be happy to review my score as a result of the rebuttal process if my points are properly tackled. I number my comments for easier discussion.


1. A key weakness of this work is that it seems to me that there's an overstatement when the authors write that this is the first work to propose a solution to the issue of strutural-functional model regional heterogeneity. Maybe this issue is ill-defined and need a more clear description, otherwise I don't see how some of the previous works mentioned do not try to create models that take into account different functional-structural interactions on different regions of the brain. For example, work [59] (mentioned in introduction as an example of "self-attention" technique) seems to use attention layers at different layers of the model and due to its (attention) nature, it seems too much to call it a "simple pattern". Similarly, for work [58], I believe the authors are oversimplifying it by classifying it as just a "weighted summation", as it seems to me to be a much more complex usage of graph neural networks (GNNs) to fuse in a non-linear fashion different modalities of the brain. Indeed, work [58] seems to follow other works that I've seen leveraging GNNs to fuse different brain modalities (eg, 10.48550/arXiv.2007.09777, 10.1016/j.media.2022.102471, 10.1007/978-3-030-32248-9_89) which the paper did not mention and for me they are clearly tackling the issue of funtional-structural interactions even if they did not use this exact name. Finally, even for the simplest case of just concatenating (embedded) features it doesn't mean a model is making a "one-to-one mapping" as defended in the paper.
2. A second key weakness I see in this paper is how some decisions seem to be justified/made in section 3.3.1. Section 3.3.1 starts with "Due to the issue of regional heterogeneity between SC and FC, it is obviously not feasible to pass messages directly between two modalities". What do the authors mean by "messages" here and why is this so obvious? Wouldn't "messages" between the different modalities actually help in the issue of not just concatenate two modalities? This seems to be related to lines 187-189, when the paper says that it avoids direct interactions between modalities. Why is direct communication such a bad thing and even called a "forbidden" interaction as indicated in figure 3? It seems to me that direct communication, together with other means of "communication" could help any model achieve better performance. Could maybe the authors justify this point a bit better? A possible ablation analysis I see here to support this statement would be to remove Z_b altogether, and this would imply that just one transformer would be required to fuse functional/structural data, greatly reducing the complexity of the model for (maybe) a similar performance.
3. The final key weakness I have to identify in this paper is that it's not clear to me whether the results are overly-optimistic, and I will appreciate any clarification the authors can provide on this if I misunderstood this point. The paper seems to use a single train/validation set (for each fold of the 10-fold CV procedure), therefore the evaluations in table 1 seem to be made on the validation set. It also seems the case that this same validation set is used to know when to stop the training procedure. If I understood this correctly, then the results in table 1 can be overly-optimistic, as the same data used in training (eg, to know when to stop) was also used for final evaluation.

(Small typo, in Line 203, should not capitalise "We")

**Questions:**

Beyond what I wrote in the "Weaknesses" section, I have a few more questions:

1. Am I correct to understand that both the functional and structural modality need to be parcellated with the same atlas? In figure 1's representation I got the impression that the two graphs could be different, but then from the text it doesn't seem that this is the case. Can the authors please clarify this?
2. Have the authors considered analysing different choices for the readout function to check for possible result improvements?
3. Have the authors considered other tasks beyond binary classification? For example a regression-based task like age prediction?
4. Around line 249, if the feature matrix of the structural data is of shape |V_{SC}| x |V_{SC}|, doesn't it mean each node will have repeated features given the symmetric nature of the matrix?
5. Around lines 259/260, I understood that the data is split in 50% train data and 50% validation data. If this is true, can the authors clarify why this was the case? In our field it is more common to have a higher percentage for the train data, and that's why I ask this.
6. Although it is good that the paper contains traditional ML models as baselines (ie, SVM, random forest), why didn't the authors choose the traditional methods mentioned in the Related Work section?
7. Work [59] seems to be the only previous work mentioned by the authors that use self attention. Given the model proposed by the authors highly relies on Transformers, shouldn't this work be included in the baselines comparison?
8. Do the authors have any hypothesis on why the results of the two-site brings worse results than the two datasets separately? Could it be the way the two sites are separated when splitting the data in train/val sets? (I am assuming more data usually means better generability)

**Limitations:**

The paper highlights some limitations, but no potential negative societal impact of work seems to have been mentioned. The fact that the model uses medical data, and that it can be used to predict depression, seems to me to be potentially be used in negative ways with direct impacts in someone's life?

---

> ### Author Rebuttal · Authors · 2023-08-03
>
> First of all, thank you for your recognition of our work, but for the disclosure of data, it does need to be followed up by our discussion with multiple parties (including but not limited to the hospital side).
>
> Answers to weaknesses：
> 1. Maybe we did ill-define this issue, and the lack of clarity in the description. The reason why we say this is the first work is that our work focuses on indirect interactions of regional heterogeneous modalities rather than direct interactions, and we really did not find any work done in this field. As for the later examples of work [58, 59], they did do a lot of meaningful research on multimodal fusion, but they are still essentially direct fusion of multimodal information, so we define them as simple pattern (we define direct interaction as a simple pattern, not that these works are simple).
>
>    Secondly, "one-to-one mapping" just describes the fact that there is no "one-to-one mapping" between function and structure, not used to describe any previous work.
>
> 2. The "message" is simply understood as the semantic information within the modality (like node's feature or the graph's structure information, etc.).
>
>    Secondly, we do not say that the information from different modalities is detrimental to the final task, but we emphasise that our research focuses on heterogeneity research, and the indirect interaction between different modalities, so the information from different modalities is still interacting in essence, but we have converted the direct to indirect.
>
>    Thirdly, to address this "bad thing" problem, because of the heterogeneity problem, we conceived the framework of indirect interaction, and finally verified the role of indirect interaction through experiments, so we believe that direct interaction is a bad thing compared to indirect interaction, so "forbidden" interactions mean that direct interactions between different modes are removed from our model. We did the corresponding ablation experiment in the paper, which may have been misunderstood because we did not describe it clearly. As in the ablation experiment, w/o Trans-Bottlenecks is to remove the fusion bottleneck, and use a standard Transformer to directly interact with the two modal information (which is what you want for ablation analysis).
> 3. We can understand your concern. Our experimental setup is indeed using a 10-fold CV procedure with only the training and validation sets divided. The reason why we didn't divide the validation set and test set separately is that we researched a lot of articles and found that a lot of brain science related studies used cross-validation (refer to 10.1016/j.media.2021.102082, 10.1109/TNNLS.2022.3154755, 10.1016/j.media.2022.102550, 10.1109/TMI.2022.3222093), so we also adopted this approach. In our opinion, this 10-fold CV procedure has conducted experiments on multiple divisions and can better overcome the effects of division randomness.
>
> Answers to questions:
>
> 1. Structure and function are really two different graphs, they are two graphs with no connection at all, two modal information. The same atlas refers to the fact that we used the same atlas brain partitioning template, so that the two modal graphs have the same number of nodes and nothing else is the same.
> 2. Thanks to your suggestion, we now add this experiment in the "global" rebuttal pdf file (refer to Tab. 1 and Tab. 2 in the "global" rebuttal pdf).
> 3. It is true that we did not consider other types of tasks in the paper because our current study is based on current clinical needs, and in the future if there is a need for other types of tasks, we believe that our model can do a good job of migrating as well.
> 4. Because SC essentially represents the number of fibre connections each brain region has to other brain regions, the feature matrix is an is of shape |V_{SC}| x |V_{SC}|. There may exist some of the same values, but they have different meanings.
> 5. As we explained in detail in the answer to the third weakness above. In each fold, 90% of the data is used for training and 10% of the data is used for validation and testing.
> 6. In this paper, we focus more on machine learning (including deep learning) methods, so we do not compare with SNF or ICA, which are traditional methods, in our comparison experiments.
> 7. The fact that work [59] is based on brain imaging, whereas our study is based on brain network graphs, there is a fundamental difference between the image and graph. So we didn't include it in the comparison experiments (as it is mentioned in the paper in line 273-274 that our comparison methods are selected to be the research methods that are directly on SC and FC). The second thing is that work [59] does not disclose their code. We eventually added AL-NEGAT as a comparison method, which is an attentional-based approach.
> 8. Since multi-site data are collected from different scanners with different acquisition parameters, non-neural inter-site variability may mask inter-group differences. Although multi-site increases the data but the overall data distribution is more complex, so it leads to performance degradation. In the experiments of two-site, we first divide the two datasets individually (90% train, 10% validation), and then combine them into a complete two-site dataset. Some multi-site work also leads to performance degradation, such as 10.1016/j.media.2021.102279, 10.1016/j.media.2020.101765.
>
> Answers to limitations:
>
> Regarding the negative social impacts mentioned, firstly, our research is currently limited to scientific studies and has not been put to industrial use. Secondly, our data were collected with the consent of the subjects who were informed of the purpose for which the samples were collected. We also observe ethical and moral principles, and all of our samples were collected with personally identifiable information hidden. Overall, we can guarantee that the present exercise will not bring about any negative social impact.

---

> > ### Comment · Reviewer_7Tgh · 2023-08-11
> >
> > I thank the authors for the time spent answering my concerns point by point. I have some follow-up comments and questions, which I'll do point by point as well.
> >
> > Regarding the weaknesses:
> > 1. It seems to me what the authors are trying to defend is still an over-statement. In the rebuttal to this point, the authors try to explain something that we agreed might seem to be ill-defined, by using two concepts (direct interactions and indirect interactions) without exactly explaining them in detail. From what I understand as indirect and direct interactions, I still don't see how how previous works ([58, 59], and the ones I mentioned) cannot be seen as exploiting indirect interactions. We are talking about previous works that use attention, and leverage complex interactions using GNNs. For GNNs in particular, they are able to exploit various hops of information, and therefore I cannot see how these previous works are exploring only direct connections or "one-to-one mappings" between function and structure.
> >
> > 2. I think I understand the ablation analysis better now, thanks a lot for the clarification. I see now that the ablation "w/o Trans-Bottleneck" actually corresponds to the ablation analysis I suggested, so I apologise for my confusion. I also see how this result tries to show that, on average (as per table 2), having that fusion bottleneck "forbidding" direct interactions seems to show that direct interactions are not as good as indirect interactions. In this case, the key weakness of this point is a bit different from what I initially understood. This specific ablation result does not seem to be significantly different (in table 2), with averages sometimes very close to each other and with big/overlapping standard deviations. I'd say therefore that what seems to be happening here is that the increase in averaged performance actually happens because we have a significant more complex model (two transformers) instead of just one transformer, and that the overlapping standard deviations seem to show that results are so similar that the proposed model in the paper might not be worth double the cost to bring a marginal improvement. Thus, this experiment actually seems to show that direct connections are so important that a single transformer is able to leverage enough information to get similar performance at half the computation cost of the fusion bottleneck that the authors propose. This also seems to show to me that the sentence "Due to the issue of regional heterogeneity between SC and FC, it is obviously not feasible to pass messages directly between two modalities" is not properly defended as the wording is too strong (ie, *obviously* not feasible) for the results presented. In this rebuttal point the authors seem to agree with me that information from different modalities can be useful, they just decided to focus on this more indirect connections given their results. However, as I try to defend here, this doesn't seem that *obvious*.
> >
> > 3. The authors seem to agree with me that their CV approach uses the same split for the validation and test set. In this case, then, this is still a key weakness and results might be overly-optimistic. I am well aware, as the authors point out, that a lot of ML literature has a serious issue with data leakage and wrong evaluation processes. However, for a conference like neurips, we'd expect good/accepted papers to be aware that if a training procedure uses a validation set either for hyperparameter search or to know when to stop, an independent test set is required to report final performance. I agree 100% with the authors that a CV procedure is a good - and actually necessary - procedure to report results. The main issue, though, is that the test sets of this CV procedure match the validation set used to train the neural networks at each split, and that is wrong.
> >
> >
> > With regards to my questions, I just have follow-up questions/comments to two of them:
> > 1. If the graphs of the two modalities have the same number of nodes, then the authors might want to consider updating figure 1 for increased readability. As I said, it seems to show that the two modalities might have a different number of nodes.
> > 6. Those other methods that the authors mention in the Related Work section, like SNF and ICA, seem to be important in the context of multimodal brain fusions. In this sense, I do not understand why the authors preferred to focus only on more traditional ML models. SVM and random forests are traditional ML models that are not based on deep learning, and therefore contribute to show the relevance of this work, even though they didn't seem to have been relevant in the context of the Related Work section. Therefore, it seems even more important that the authors use SNF and ICA as baselines to show the relevance of their work in the context of the literature that the authors present in the Related Work section.

---

> > > ### Author Response · Authors · 2023-08-16
> > >
> > > Answers to weaknesses:
> > > 1. We apologise for not explaining direct and indirect interactions in detail in our previous rebuttal, and therefore we re-explain both concepts here.
> > >
> > >    Regarding direct interaction, we mean two modalities (i.e., two graphs) whose features/embeddings are put together to do some computation, whether it is a weighted sum of the feature matrices or a potential embedding computation using attentions on them.
> > >
> > >    We define indirect interaction as the opposite of direct interaction, where the features/embeddings of the two graphs are not put together to do some computation, but they are put together with the bottleneck respectively, i.e., the bottleneck serves as a bridge for the interaction of the two graphs.
> > >
> > >    Starting from the above definition, the previous work [58][59] and what you mentioned, they both put the features/embeddings of the two graphs together to do certain computations, and therefore fall into this category of direct interaction.
> > >
> > >    Regarding your statement that GNNs can pass information in multiple hops, you may have misunderstood the concept of direct and indirect interactions, the multi-hop passing of information in GNNs is within a graph and between multiple nodes, whereas the direct and indirect interactions that we have defined are for two graphs, which is a completely different thing.
> > >
> > > 2. In response to the second weakness, firstly, thank you for your detailed explanation, our previous wording does need to be reconsidered. For your question about the effect not being obvious, we think that indirect interaction does bring some performance improvement from the experimental results. Our starting point at the very beginning of this paper was also to explore the role of indirect interactions, which is an unexplored issue, and thus we focused more on indirect interactions. But as you said (and as we agreed in our previous rebuttal), direct interaction must be beneficial for multimodal tasks, and so in future work we will combine direct and indirect interaction for further exploration.
> > >
> > > 3. In response to the third weakness, thank you for your suggestion, we recognise your statement and therefore we have redesigned our experimental evaluation process. We still use 10-fold cross validation, but in each fold, we divide the dataset into 8:1:1 (80% for train, 10% for validation and 10% for test). Finally, the average results of 10-fold cross-validation are counted. Due to the current time constraints, we only give the new comparison experimental results of the HCP dataset in the rebuttal for the time being, and the other new results will be given in a future version of the paper.
> > >
> > > |         |  | **HCP Dataset** |          |           |            |         |
> > > |:-----------------:|:------------:|:---------------:|:--------------:|:--------------:|:--------------:|:------------:|
> > > |       **Method**        |      **Modality**        | **ACC**             | **SEN**            | **SPE**           | **F1**             | **AUC**         |
> > > | **FGDN**          | fMRI         |  64\.87±2\.76   | 59\.28±12\.18  | 69\.64±9\.90   | 60\.81±5\.05   | 64\.46±2\.99 |
> > > | **FGDN**          | sMRI         |  60\.26±3\.25   | 51\.25±18\.56  | 68\.00±13\.33  | 52\.48±12\.16  | 61\.62±3\.94 |
> > > | **BrainGNN**      | fMRI         |  63\.62±4\.12   | 64\.52±6\.52   | 61\.87±11\.21  | 60\.32±4\.98   | 61\.78±5\.97 |
> > > | **BrainGNN**      | sMRI         |  64\.01±3\.99   | 65\.00±5\.99   | 62\.24±9\.76   | 61\.53±4\.74   | 63\.65±5\.80 |
> > > | **SVM**           | fMRI, sMRI   | 62\.27±3\.25    | 53\.25±18\.56  | 70\.00±13\.33  | 54\.48±12\.16  | 61\.62±3\.94 |
> > > | **Random Forest** | fMRI, sMRI   | 66\.51±2\.67    | 52\.41±6\.32   | 78\.57±4\.79   | 58\.89±4\.38   | 65\.49±2\.76 |
> > > | **SNF**           | fMRI, sMRI   | 53\.89±4\.96    | 51\.56±10\.21  | 55\.21±9\.22   | 52\.14±10\.83  | 59\.25±6\.38 |
> > > | **MGCN**          | fMRI, sMRI   | 66\.85±4\.95    | 60\.33±9\.20   | 74\.28±6\.64   | 63\.13±6\.39   | 67\.31±5\.09 |
> > > | **GBDM**          | fMRI, sMRI   | 65\.05±5\.23    | 61\.80±6\.22   | 67\.55±9\.45   | 62\.61±4\.65   | 64\.68±5\.02 |
> > > | **MMGNN**         | fMRI, sMRI   | 67\.46±4\.27    | 53\.03±14\.48  | 79\.82±8\.10   | 58\.76±11\.65  | 66\.42±4\.81 |
> > > | **AL\-NEGAT**     | fMRI, sMRI   | 68\.19±3\.11    | 64\.71±7\.15   | 72\.18±4\.75   | 66\.38±5\.12   | 68\.94±4\.23 |
> > > | **Ours**          | fMRI, sMRI   | **71\.51±4\.53**    | **66\.76±10\.05**  | **80\.82±4\.67**   | **69\.01±6\.92**  | **72\.79±4\.86** |
> > >
> > >
> > > Answers to questions:
> > > 1. Thank you for your careful observation, we will update figure 1 in the new version of the paper.
> > > 2. Thanks to your suggestion, we added SNF as the baseline method in the new comparison experiment, as for ICA, we failed to find a public implementation of it, so it was not included in the comparison experiment.

---

> > > > ### Comment · Reviewer_7Tgh · 2023-08-21
> > > >
> > > > I thank the authors for their answers to my remaining points.
> > > >
> > > > With regards to my weakness number 1, I believe the authors and I are in disagreement with regards to the novelty of this work, and it really still seems to me there's an overstatement when the authors write that this is the first work to propose a solution to the issue of strutural-functional model regional heterogeneity. Under this framework of direct/indirect interactions, just because there's not a dedicated bottleneck in the middle of the two modalities it doesn't mean the other architectures cannot capture some complex indirect patterns. Therefore, I still think that the usage of the terms "weighted sum of the feature matrices" and "potential embedding computation using attentions on them" is an oversimplification of what these previous works are doing.
> > > >
> > > > From our further discussions in weakness 2, I think we are still in disagreement with regards to the significance of this work as  I'd still say that what seems to be happening here is that the increase in averaged performance actually happens because we have a significant more complex model (two transformers) instead of just one transformer, and that the overlapping standard deviations seem to show that results are so similar that the proposed model in the paper might not be worth double the cost to bring a marginal improvement.
> > > >
> > > >
> > > > For the remaining points I had, the authors did a very good job in addressing them, and therefore I'm increasing my decision from borderline accept to weak accept, the soundness score from 2 to 3, and the presentation score from 3 to 4.

---

> > > > > ### Author Response · Authors · 2023-08-21
> > > > >
> > > > > We would like to thank you for recognising our work and for raising our score.
> > > > > We will further improve the presentation in the revised version.
> > > > >
> > > > > Thank you again for your meaningful comments.

---

### Official Review · Reviewer_A4mz · 2023-07-21

**Soundness:** 2 fair
**Presentation:** 3 good
**Contribution:** 2 fair
**Rating:** 5
**Confidence:** 4

**Summary:**

The author proposes a novel regional heterogeneous multimodal brain networks fusion strategy to alleviate the issue of regional heterogeneity of multimodal brain networks. This strategy uses a graph convolutional network for the extraction of initial features of nodes (brain region from AAL atlas) and a transformer-based fusion bottleneck module for the fusion of structural connectivity matrix and functional connectivity matrix. This fusion strategy achieved the best results on the tasks of gender classification and depression classification.

**Strengths:**

    1. This work is well written and the figure is clear, making it easy for the reader to understand what has been done.
    2. This work is of great significance. How the structural and functional networks of the brain fusion have been an open question because the understanding of how the anatomical constraint is related to the elaborate functions is nevertheless fragmentary. Addressing the fusion of brain structure and function from a deep learning perspective can help facilitate understanding of the relationship between structure and function.

**Weaknesses:**

    1. The authors state "alleviate the regional heterogeneity between multimodal brain networks". This is a very ambitious question, and the high classification accuracy alone does not account for the alleviation of inter-regional heterogeneity. The authors should do detailed analysis experiments based on brain regions on the basis of high accuracy to explore which brain regions are non-heterogeneous between modalities, which brain regions are heterogeneous between modalities, and whether removing these heterogeneous brain regions can achieve better classification results.
    2. The authors state "they were inspired by MBT, so how does your work differ from MBT"? It seems that the author's thinking is to not allow interaction of tokens between structure and function, why is that, shouldn't structure and function be meant to interact with each other? The authors say that it effectively improves the performance of the model (line 189), but I can't see from the experiments that there are results to prove this (Transformer with bottleneck token rather than w/o Trans-Bottleneck). The authors do not compare it with MBT's approach to prove the effectiveness of their proposed method.
    3. The authors state in line 257 that the initial features of the function (Xfc) are averaged time series, but the Xfc shown in Figure 1 is the function connectivity matrix. These two points are contradictory. And, the initial features of the structure are the structural connectivity matrix, so why should the initial features of the function be the time series of fMRI instead of the functional connectivity matrix?
    4. Are the Final Bottlenecks obtained based on Zb only? If so, why not cascade Zsc, Zb, Zfc as inputs to the MPL layer? The authors need to do ablation studies to prove that Zb-based classification is the most accurate. What does Nb in Equation 4 mean?
    5. How is the threshold taken for the connectivity matrix to the adjacency matrix for the two modes? Firstly, the authors didn't write how much the threshold is taken. Secondly, the choice of the threshold for the structural and functional connectivity matrices has a big impact on the downstream task, it is suggested to do ablation studies to show how you take the threshold.

**Questions:**

See Weaknesses for details.

**Limitations:**

See Weaknesses for details

---

> ### Author Rebuttal · Authors · 2023-08-04
>
> Answers to weaknesses：
>
> 1. For the first weakness, we can understand your concern, and we have thought about it in the same way, but we have not come up with a good way to reflect the regional heterogeneity of brain networks at this time. Your suggestion of exploratory experiments based on brain regions was something we had considered, but since the number of brain regions present in each of our graphs is 90, it would require 2^90 experiments if we were to conduct exploratory experiments on brain regions, which is clearly unrealistic. So for now, we are still only able to measure the model in terms of classification performance on the downstream task.
>
> 2. For the second weakness, since this work is investigating the regional heterogeneity between SC and FC, we believe that direct interaction will not achieve better results due to misaligned feature embeddings. So this indirect interaction of fusion bottlenecks is proposed.
>    The focus of our approach is that structure and function do not interact directly, but it does not mean that they do not interact. We adopt an indirect interaction approach, which is Figure 3 in the paper, where the structure and function interact indirectly with each other through fusion bottleneck.
>
>    Secondly, about the performance improvement of fusion bottleneck on the model. Actually, there is an experimental proof in the paper, maybe we didn't describe it clearly enough. The experimental setup about w/o Trans-Bottleneck is actually mentioned in the paper in line 308, which is to use the standard Transformer to directly compute the tokens of the two modalities, so for the third and fourth rows in Table 2, this ablation experiment actually shows the role of the Fusion bottleneck (Fig. 3) (i.e., the comparison of the difference between direct and indirect interactions).
>
>    Finally, about not comparing with MBT, it is because MBT itself is in the field of computer vision, doing audio-video modal fusion, so it can't be used directly in our scenarios, so I didn't include it in the comparison experiments to compare the methods (because it is mentioned in the paper in line 273-274 that our comparison methods are selected to be the research methods that are directly on SC and FC).
>
> 3. For the third weakness, this one is indeed our writing error, what is shown in Figure 1 is the correct one, using the functional connectivity matrix as a feature.
>
> 4. For the fourth weakness, thank you for this good suggestion, we are attaching the additional experiments to the rebuttal.
>
>    We apologize that Nb stands for the number of Bottlenecks, which is not specifically labelled here because it was mentioned earlier.
>
> 5. For the fifth weakness, regarding the value of the threshold, in fact, we have done an ablation experiment on the threshold before, through which we finally selected the optimal value as the threshold for the subsequent experiments. However, we did not put the results of this ablation experiment into the paper because we thought that this was not the focus of our research, and now we give the results in the "global" rebuttal pdf file (refer to Fig. 1 in "global" rebuttal pdf).
>
> | Input                  | HCP Dataset  |              |              |              |              |
> |------------------------|--------------|--------------|--------------|--------------|--------------|
> |                        | Acc          | Sen          | Spe          | F1           | Auc          |
> | **Z\_sc \+ Z\_fc**         | 77\.62±3\.69 | 74\.96±7\.70 | **81\.79±7\.40** | 76\.30±4\.33 | 78\.38±3\.68 |
> | **Z\_sc \+ Z\_b \+ Z\_fc** | 77\.57±4\.06 | 74\.77±9\.26 | 80\.00±9\.03 | 75\.34±4\.50 | 77\.38±4\.03 |
> | **Only Z\_b \(ours\)**     | **78\.63±4\.36** | **75\.59±6\.75** | 81\.25±6\.04 | **76\.49±4\.91** | **78\.42±4\.38** |
>
>
> | Input                  | Two\-site Dataset |              |              |              |              |
> |------------------------|-------------------|--------------|--------------|--------------|--------------|
> |                        | Acc               | Sen          | Spe          | F1           | Auc          |
> | **Z\_sc \+ Z\_fc**         | 75\.73±1\.83      | 69\.01±4\.87 | **82\.39±3\.01** | 72\.55±3\.04 | 75\.70±1\.82 |
> | **Z\_sc \+ Z\_b \+ Z\_fc** | 75\.29±1\.89      | 69\.09±3\.96 | 81\.50±4\.89 | 72\.34±2\.58 | 75\.29±1\.89 |
> | **Only Z\_b \(ours\)**      | **78\.48±1\.43**      | **76\.20±4\.06** | 80\.72±3\.60 | **77\.35±1\.97** | **78\.46±1\.43** |

---

> ### Comment · Reviewer_A4mz · 2023-08-12
>
> Thank the authors for the rebuttal. All my raised questions have been properly solved. Hence, I would like to revise the rating to borderline accept.

---

> > ### Comment · Reviewer_7Tgh · 2023-08-13
> >
> > I need to ask the reviewer A4mz: if all their questions raised have been properly solved, why isn't the reviewer giving a higher scorer? Even more because it's asked that reviewers use the "borderline" decisions "sparingly". I believe this is important for transparency with the authors, for an easier decision by the other reviewers and area chairs.

---

> > > ### Comment · Reviewer_A4mz · 2023-08-18
> > >
> > > I raised the rating because the authors solved my questions. I did not give a higher score because of the novelty. The novelty of the proposed multimodal fusion framework and transformer-based fusion bottleneck module is insufficient. They merely modified the direct interaction of MBT into an indirect one. The authors stated that their work can alleviate the regional heterogeneity between multimodal brain networks. The final results demonstrated that a network combining two modalities through a certain framework performs better classification. However, they did not delve into brain region-based analysis or explanatory analysis to demonstrate how this fusion approach alleviates the heterogeneity of multimodal brain networks.

---

> > > > ### Author Response · Authors · 2023-08-20
> > > >
> > > > Firstly we are grateful to the reviewer A4mz for raising our score. We appreciate your time and effort in providing us with insights to improve our work. We have carefully considered your point about the brain region-based analysis, and we acknowledge its significance in providing a comprehensive understanding of our research.
> > > >
> > > > While we recognize the importance of conducting a deeper brain region-based analysis, we also want to be transparent about the current limitations of our study. Given the scope and resources available, a detailed brain region-based analysis exploring the intricate nuances of brain region-based interactions might be beyond the recent capacity of this study.
> > > >
> > > > However, we believe that your suggestion is highly valuable. In future work, we will make a dedicated effort to delve into the brain region-based analysis aspects you've highlighted. This will enable us to provide a more robust and thorough analysis of how our fusion approach addresses regional heterogeneity within multimodal brain networks.
> > > >
> > > > Thank you again for your insightful comments.
> > > >
> > > > Regarding the innovative question, we would like to restate the innovative of this work.
> > > >
> > > > An important step in the method proposed in this paper is indeed from direct interaction to indirect interaction, but our focus is:
> > > >   1. raise a never-before-explored issue (i.e. using indirect interaction to alleviate the heterogeneity issue of multimodal brain networks).
> > > >   2. the use of brain graph networks to extract local features.
> > > >   3. the use of fusion bottleneck for indirect fusion of multimodalities. the use of fusion bottleneck for indirect fusion of multimodalities.

---

### Official Review · Reviewer_8rTt · 2023-07-25

**Soundness:** 3 good
**Presentation:** 3 good
**Contribution:** 2 fair
**Rating:** 7
**Confidence:** 5

**Summary:**

The article discusses the use of multimodal fusion as a research technique in neuroscience to extract complementary information from multiple modalities. Since previous research has neglected the regional heterogeneity between structural connectivity (SC) and functional connectivity (FC) and used inefficient ways of multimodal fusion. To address this issue, the authors proposed a novel Regional Heterogeneous multimodal Brain networks Fusion Strategy (RH-BrainFS), which uses a brain subgraph networks module to extract regional characteristics and a transformer-based fusion bottleneck module to alleviate the regional heterogeneity between SC and FC. The proposed method outperforms several state-of-the-art methods in various neuroscience tasks and is the first work to propose a solution to the issue of structural-functional modal regional heterogeneity.

**Strengths:**

First the research problem is significant. The structural-functional modal regional heterogeneity is a recent popular research topic. The authors first propose a method which could work well in this situation. The novelty of the paper is sufficient. To alleviate the issue of regional heterogeneity of multimodal brain networks, the authors propose a novel Regional Heterogeneous multimodal Brain networks Fusion Strategy (RH-BrainFS), using BrainSubGNN module and Trans-Bottleneck module to fuse regional heterogeneous multimodal brain networks for neuroscience tasks.The experiments are sufficient, the authors conduct the experiment on two main-stream MRI brain benchmarks. Extensive experiments demonstrate the effectiveness of RH-BrainFS in multimodal brain networks fusion tasks on depression classification and gender classification datasets.

**Weaknesses:**

1. authors have not explained in detail why using "forbidden Interaction" in the Fusion bottlenecks. Authors are suggested to provide more motivations and explanation of this part.
2. authors have not provide the detailed speed and time-consumption comparison with other methods. The proposed method seems time-consuming.

**Questions:**

1. the authors use thresholding to get Asc and Afc from Xsc and Xfc, how the selection of thresholding effect the final results? how sensitive the results are.

**Limitations:**

see the weakness part

---

> ### Author Rebuttal · Authors · 2023-08-04
>
> Answers to weaknesses：
> 1. For the first weakness, "forbidden interactions" means that we remove direct interactions between SC and FC (just as in the standard Transformer model, a direct calculation of attention scores for different modal markers is a direct interaction). The initial motivation for this was that we believed that there was regional heterogeneity between SC and FC, and thus we believed that their features or potential embeddings would be misaligned, and thus direct interaction would lead to poorer results. Therefore, we proposed indirect interactions for fusion bottlenecks, and later also experimentally verified the effectiveness of indirect interactions (compare the ablation experiments in the third and fourth rows of Table 2). Therefore, we removed the direct interaction between SC and FC (i.e. direct interaction is forbidden) and adopted the indirect interaction of the fusion bottleneck.
>
> 2. For the second weakness, we supplemented the time-consumption experiment (record the average training time for 100 epoches and the average time for 1 inference on the Two-site dataset), and as you would expect, our method incurred a greater time overhead.
>
>    To explain such results, we believe that firstly MGCN, GBDM and MMGNN are more focused on researching on processing of data features, so common GNN frameworks (which are simpler compared to Transformer or Attention) are used in the network structure, so they have better good computational efficiency.
>
>    Secondly, we consider AL-NEGAT to be an attentional-based approach that has similarities with our Transformer-based, so AL-NEGAT generates a time overhead that belongs to the same order of magnitude as Ours.
>
>    In response to such results, we admit that our method does underperform in terms of computational efficiency, but falls within an acceptable range, after all, our model performance is better improved.
>
> Answers to questions：
> 1. For the first question, regarding the value of the threshold, in fact, we have done an ablation experiment on the threshold before, through which we finally selected the optimal value as the threshold for the subsequent experiments. However, we did not put the results of this ablation experiment into the paper because we thought that this was not the focus of our research, and now we give the results in the "global" rebuttal pdf file (refer to Fig. 1 in "global" rebuttal pdf).
>
> | **Method**    | **Train Tine \(s / 100 epoches\)** | **Inference Time Cost \(s / 1 inference\)** |
> |:-------------:|:-----------------------------------:|:--------------------------------------------:|
> | **MGCN**      | 23\.875                           | 0\.017                                     |
> | **GBDM**      | 20\.388                          | 0\.013                                     |
> | **MMGNN**     | 23\.905                          | 0\.016                                     |
> | **AL\-NEGAT** | 31\.573                         | 0\.023                                     |
> | **Ours**      | 33\.875                          | 0\.025                                    |

---

> > ### Comment · Reviewer_8rTt · 2023-08-19
> >
> > After the rebuttal, authors well addressed my concern. So I raised my rate to 'accept'.

---

> > > ### Author Response · Authors · 2023-08-21
> > >
> > > Thank you for recognising our work and raising your score.

---

### Author Rebuttal · Authors · 2023-08-05

In response to some reviewers and ethics reviewer's questions about the negative social impact, we would like to explain the following.

First, regarding the dataset collection process, the Human Connectome Project (HCP) dataset, as a publicly available dataset that has been used in numerous previous studies, is undoubtedly not ethically questionable. It is true that the hospital dataset is held in collaboration with our partner hospitals and is not yet publicly available, but the data is collected with the consent of the subjects who are clearly informed of the purpose of the sample collection, and all identifying information about the sample is hidden in the hospital dataset. Therefore, it does not adversely affect any individual, so there are no ethical or moral issues.

Secondly, with regard to research on depression, the work we have done so far has been limited to the scientific research stage and has not been put to clinical application. We can guarantee that all the research conducted in this work will not have any negative social impact.

Thirdly, regarding the ethics reviewer's question about the dataset being too small and the use of the depression dataset. In fact, this is really the only dataset that available to us, because there are great difficulties in data collection in this field, and also, processing the data is very costly, so it is really small for the time being. For the question of using depression datasets, the previous two points have basically answered the question without any moral or ethical issues, and it is currently limited to scientific research without any negative social impacts.

(The explanation above will be appended to the full paper of the new version.)

In this global rebuttal we accompany the results of some supplementary experiments (a pdf file).

---

> ### Comment · Reviewer_7Tgh · 2023-08-11
>
> I find it very worrying that the authors state in such a strong tone that they can **guarantee** that the research conducted in this work will not have any negative social impact. This is by definition false, due to the simple fact that no researcher can control how other researchers or the general/specialised people might use their work. Therefore, even the "best" work could have a "bad" side. I believe this is a strong reason for neurips to ask authors to include a negative social impact section, such that we try to be more aware of possible risks, even if we cannot control everything. As I said in my review, the fact that the model uses medical data, and that it can be used to predict depression, seems to me to be potentially be used in negative ways with direct impacts in someone's life. What if the model wrongly predicts that someone has depression and that a doctor agrees with this outcome? That same person could be prescribed drugs unnecessarily with impacts to their health and the economy and society. And this is just one example. Just because one possible negative outcome seems unlikely, it doesn't mean it's not a potential one.
>
> From the authors' response, I believe they are confusing ethics/moral issues with **potential** negative impacts in society. I agree with the authors that their work do not seem to raise any ethical flag, and I hope that reviewer UEYi can explain why they thought so in more detail. However, that doesn't mean that the work couldn't potentially have negative social impacts in society, and thus I urge the authors to reconsider their claims in this context.

---

> > ### Author Response · Authors · 2023-08-12
> >
> > Thank you for the detailed explanation, we probably did confuse negative social impacts with ethical issues in our previous answer.
> >
> > Regarding the issue of negative societal impact, we would like to re-explain it here.
> >
> >   1. Firstly, as stated in the rebuttal, our research is currently limited to scientific research, not for practical use, and will be in such a stage for a long time.
> >
> >   2. Secondly, the use of AI methods in the study of medical diseases is an important area of research at the moment, and it is extremely important to promote in-depth research in this area, and there may be a potential negative social issue with most research in this area, and I agree with your point that "we can't control the other researchers", and therefore only in our study, we believe that even if it is put to practical use in the future, it will only be used as a medical aid to assist doctors in diagnosis and will not be the decision makers. The purpose of the tool is to save doctors' time and effort in order to complete the diagnosis of more patients.
> >
> > (About the negative social impacts we are currently thinking about, we will give it in another thread.)

---

> ### Author Response · Authors · 2023-08-12
> **Negative social impact**
>
> We thought about the possible negative social impacts of this study, which we list here as well as explain as necessary (this section will be added in a future new version of the paper).
>
> 1. **Incorrect diagnosis:** AI methods must have the possibility of error, which cannot be avoided, but an incorrect diagnosis will have a significant impact on individuals and society. Therefore, AI tools can only be used as a diagnostic aid, not as a decision maker, and the final decision should still be made by the doctor.
> 2. **Leakage of privacy information:** In depression dataset, the identity information of the subjects is highly private, and the leakage of identity information will also have unpredictable and significant impact on individuals and society. Therefore, in this work, we have completely hidden the subjects' identifying information (which is also not visible to the staff in the study group) as a way of preventing the leakage of private information.

---

> > ### Comment · Reviewer_7Tgh · 2023-08-12
> >
> > I thank the authors for further thinking on potential negative impacts and stating they will include the ones they thought in a future version of the paper. I also agree with them that it's important to promote in-depth research in this area, and that most of this research may have potential negative social issues.

---

### Decision · Program_Chairs · 2023-09-21

**Decision:**

Accept (poster)

**Comment:**

This paper received mixed ratings, and the authors submitted rebuttals. The reviewers read the rebuttals and further discussed with the authors. Some reviewers' concerns were addressed by the authors' rebuttals and responses. As a result, three reviewers raised their ratings. The AC read the authors' rebuttals and the discussions between authors and reviewers. During the AC-reviewer discussion period, the AC further gave the following summary, which has been approved by the reviewers.

Strengths:

1) Addresses a significant research problem

2) A new spatially varying fusion strategy for multimodal Brain networks

3) Extensive experiments demonstrate the effectiveness of the proposed strategy in two multimodal brain networks fusion tasks

Weaknesses:

1) The proposed multimodal fusion framework and transformer-based fusion bottleneck module have limited novelty.

2) Improved performance at the expense of higher model complexity and computational cost

3) Lack of in-depth region-based interpretation analysis of why the proposed fusion strategy works well


At the end, the AC and reviewers feel reasons to accept outweigh reasons to reject.